# An In Vitro Catalysis of Tea Polyphenols by Polyphenol Oxidase

**DOI:** 10.3390/molecules28041722

**Published:** 2023-02-10

**Authors:** Kunyi Liu, Qiuyue Chen, Hui Luo, Ruoyu Li, Lijiao Chen, Bin Jiang, Zhengwei Liang, Teng Wang, Yan Ma, Ming Zhao

**Affiliations:** 1College of Tea Science & College of Food Science and Technology, Yunnan Agricultural University, Kunming 650201, China; 2The Key Laboratory of Medicinal Plant Biology of Yunnan Province & National-Local Joint Engineering Research Center on Gemplasm Innovation & Uilization of Chinese Medicinal Materials in Southwest China, Yunnan Agricultural University, Kunming 650201, China; 3College of Wuliangye Technology and Food Engineering & College of Modern Agriculture, Yibin Vocational and Technical College, Yibin 644003, China

**Keywords:** tea, polyphenol oxidase, tea polyphenols, metabolites, theaflavins

## Abstract

Tea polyphenol (TPs) oxidation caused by polyphenol oxidase (PPO) in manufacturing is responsible for the sensory characteristics and health function of fermented tea, therefore, this subject is rich in scientific and commercial interests. In this work, an in vitro catalysis of TPs in liquid nitrogen grinding of sun-dried green tea leaves by PPO was developed, and the changes in metabolites were analyzed by metabolomics. A total of 441 metabolites were identified in the catalyzed tea powder and control check samples, which were classified into 11 classes, including flavonoids (125 metabolites), phenolic acids (67 metabolites), and lipids (55 metabolites). The relative levels of 28 metabolites after catalysis were decreased significantly (variable importance in projection (VIP) > 1.0, *p* < 0.05, and fold change (FC) < 0.5)), while the relative levels of 45 metabolites, including theaflavin, theaflavin-3′-gallate, theaflavin-3-gallate, and theaflavin 3,3′-digallate were increased significantly (VIP > 1.0, *p* < 0.05, and FC > 2). The increase in theaflavins was associated with the polymerization of catechins catalyzed by PPO. This work provided an in vitro method for the study of the catalysis of enzymes in tea leaves.

## 1. Introduction

Tea is manufactured from the fresh leaves of *Camellia sinensis*, which is the most consumed beverage in the world after water, and widely believed to be rich in flavor compounds and have positive effects on human health, especially anti-oxidation, anti-inflammatory, gut barrier protection, and bile acid metabolism regulatory effects [1,2,3]. Polyphenols in tea leaves (TPs) account for 18% to 36% of dried tea leaves [4], mainly including catechins, *O*-glycosylated flavonols, *C*-glycosylflavones, proanthocyanidins, phenolic acids, and their derivatives, and also containing the fermented oxidation products of catechins, e.g., theaflavins, thearubigins, and theabrownins in oolong, black, and dark teas [5,6,7]. Among them, *O*-glycosylated flavonols, tannins, and galloylated catechins are the main astringent compounds, and non-galloylated catechins enhance the tea bitterness [8,9]. Furthermore, TPs are a major class of aroma compounds giving clove-like, smoky, and phenolic characteristics to dark teas, particularly Pu-erh tea [10]. Hence, TPs are important for the healthful functions and flavors of tea beverages [2,4], which have important scientific and commercial interests in tea manufacture.

According to the manufacturing process, tea can be classified into six types: green tea, white tea, black tea, yellow tea, ooloog tea, and dark tea [11]. The oxidation of TPs caused by polyphenol oxidase (PPO) or peroxidase in the manufacturing process is critical for the formation of different tea types [12,13,14]. For example, in the fixation processing of green tea, the activities of endogenous PPO and peroxidase are terminated, and TPs are not oxidized; while through fermentation, TPs are fully oxidized in black tea, and partially oxidized in oolong tea [12,15,16]. Therefore, TPs oxidation caused by PPO plays an important role in the sensory characteristics of black tea, and it has an important research value.

In the black tea production process, TPs are enzymatically oxidized by endogenous PPO to yield a complex mixture of oxidation products, including theaflavins and thearubigins, and the reaction mechanisms at the initial stages of catechin oxidation are explained by simple quinone–phenol coupling reactions [17]. Based on this, exogenous PPO can effectively increase the formation rate of epicatechin quinone and theaflavins in the solution of (−)-epicatechin and (−)-epigallocatechin [18,19] or in the solution of (−)-epicatechin gallate and (−)-epigallocatechin-3-*O*-gallate [20], increase the contents of thearubigins and theabrownins using (−)-epigallocatechin-3-*O*-gallate [21], and transform green tea extracts into black tea with a high content of theaflavins [22]. In addition, adding 1% PPO during black tea fermentation reduced the fermentation time, the content of theaflavins and thearubigins increased, and the color and aroma of the tea improved [23]. Furthermore, theabrownins, which have a healthcare function and can improve the quality of dark tea, are formed by PPO acting on TPs during the fermentation of dark tea [24,25,26,27]. However, the catalytic mechanism of PPO to TPs are complex, and effective methods are needed urgently. Therefore, in this work, to develop an effective methods for studying the catalysis of TPs by PPO, an in vitro catalysis was developed and the changes in metabolites were analyzed using metabolomic methods.

## 2. Results and Discussion

### 2.1. Optimization of Conditions for PPO Catalyzing TPs in Sun-Dried Green Tea Leaves

As shown in Figure 1A, with the increase in PPO concentrations, the contents of (−)-epigallocatechin 3-*O*-gallate (EGCG), (−)-epicatechin 3-*O*-gallate (ECG), luteolin (Lu), and gallic acid (GA) decreased significantly (*p* < 0.05), while the content of (−)-gallocatechin (GC) and (+)-catechin (C) first increased and then decreased significantly (*p* < 0.05). The content of (−)-gallocatechin gallate (GCG), (−)-catechin gallate (CG), (−)-epicatechin (EC), (−)-epigallocatechin (EGC), ellagic acid (EA), kaempferol (Kp), myricetin (My), quercetin (Qc), taxifolin (Ti), rutin (Rt), and caffeine (Ca) fluctuation changed. However, when the PPO concentrations was 500 U/mL, the contents of all TPs (e.g., GA, GC, EGC, C, Ca, EGCG, EC, GCG, ECG, Ti, CG, Rt, EA, My, Qc, Lu, Kp) decreased most significantly (*p* < 0.05) compared with control check (CK, 0 U/mL), indicating that the reaction of 500 U/mL PPO with sun-dried green tea leaves for 7 h could significantly (*p* < 0.05) catalyze TPs in sun-dried green tea leaves.

**Figure 1 molecules-28-01722-f001:**
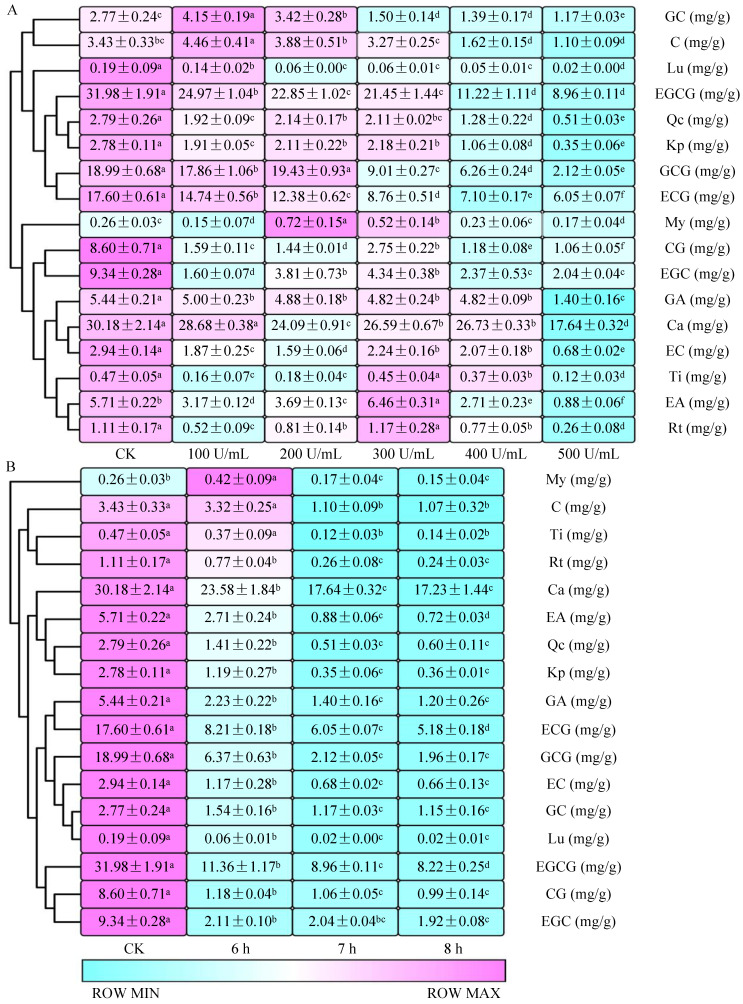
The catalysis of TPs in sun-dried green tea leaves using different concentrations (**A**) and with different reaction times of PPO (**B**). Different lowercase superscripts within a row indicated significantly different among comparisons (*p* < 0.05).

Under the action of the 500 U/mL PPO, whether the contents of TPs could be significantly (*p* < 0.05) reduced with the extension of reaction time needed further discussion, so different enzyme reaction times were selected to verify this. As shown in Figure 1B, the contents of all TPs (e.g., GA, GC, EGC, C, Ca, EGCG, EC, GCG, ECG, Ti, CG, Rt, EA, My, Qc, Lu, Kp) in sun-dried green tea leaves decreased significantly (*p* < 0.05) with the extend of enzyme reaction time from 6 h to 7 h. Furthermore, except for EGCG, ECG, and EA, the contents of most TPs did not change significantly (*p* > 0.05) at the PPO reaction time of 7 h and 8 h (Figure 1B).

Therefore, 500 U/mL PPO reaction for 7 h can catalyze most TPs (e.g., GA, GC, EGC, C, Ca, EGCG, EC, GCG, ECG, Ti, CG, Rt, EA, My, Qc, Lu, Kp) in sun-dried green tea leaves (Figure 2A). The color of sun-dried green tea leaves and tea infusions became deeper after PPO catalysis (Figure 2B). These changes in TPs were similar to those after PPO metabolism of tea leaves and are in agreement with previous reports [28,29,30].

### 2.2. Metabolomic Analysis of Catalysis of Metabolites in Sun-Dried Green Tea Leaves by PPO

Metabolomics is broadly applied in tea sciences and has tremendous potential for establishing correlations between tea metabolites and quality characteristics, and assessing the physiological changes in tea plants induced by cultivation conditions and metabolic responses to abiotic and biotic stress, and construction of metabolic pathways [31,32]. Therefore, the changes in metabolites of sun-dried green tea leaves under 500 U/mL PPO reaction for 7 h were further subjected to a metabolomic analysis. Metabolites were extracted from catalyzed tea powder (CTP) and CK, and analyzed using a non-targeted liquid chromatography electrospray ionization tandem mass spectrometry (LC-ESI-MS/MS) based metabolomics approach. As shown in Figure 3A, PCA showed that the variance contributions of PC1 and PC2 were 56.8% and 25.9%, respectively, with a cumulative variance contribution of 82.7%, which was much larger than the confidence value of 60%, suggesting that this metabolomics analysis had good stability and reproducibility. While, a great distance was observed among the three different samples, ETP samples were clustered in the upper right area, CK samples were mainly located in the bottom right, and quality control (QC) samples were clustered in the left area. Differential clustering of CK and CTP samples indicated that metabolites in sun-dried green tea leaves were significantly changed after the catalysis of PPO.

A total of 441 metabolites were identified (Figure 3B, Table 1, and Appendix A), which were classified into 11 classes, while the major metabolites included flavonoids (125 metabolites), phenolic acids (67 metabolites), and lipids (55 metabolites); these were followed by amino acids and derivatives (51 metabolites), nucleotides and derivatives (34 metabolites), organic acids (25 metabolites), alkaloids (22 metabolites), tannins (12 metabolites), lignans and coumarins (11 metabolites), terpenoids (1 metabolite), and others (38 metabolites).

They were further grouped into 33 sub-classes, including phenolic acids (67 metabolites), amino acids and derivatives (51 metabolites), flavonols (38 metabolites), nucleotides and derivatives (34 metabolites), flavones (28 metabolites), saccharides and alcohols (28 metabolites), free fatty acids (25 metabolites), organic acids (25 metabolites), flavonoid carbonoside (24 metabolites), flavanols (14 metabolites), lysophosphatidylcholine (LPC, 13 metabolites), proanthocyanidins (10 metabolites), glycerol esters (9 metabolites), alkaloids (8 metabolites), vitamins (8 metabolites), anthocyanidins (7 metabolites), lignans (7 metabolites), phenolamines (6 metabolites), lysophosphatidylethanolamines (LPE, 6 metabolites), plumeranes (6 metabolites), coumarins (4 metabolites), flavanones (4 metabolites), flavanonols (4 metabolites), isoflavones (3 metabolites), chalcones (3 metabolites), tannin (2 metabolites), and monoterpenoids (1 metabolite), etc.

To gain an overview of the differentially changed metabolites (DCMs) between the CTP and CK, we developed a new OPLS-DA of metabolites. In comparison of CTP to CK, the relative levels of 28 metabolites were decreased significantly (VIP > 1.0, *p* < 0.05, and FC < 0.5), respectively, including flavonoids (13 metabolites, e.g., quercetin 3,7-bis-*O*-β-D-glucoside, acacetin-7-*O*-β-D-glucoside, acacetin-7-*O*-galactoside, diosmetin-7-*O*-galactoside, tricin 7-*O*-hexoside, chrysoeriol, 6-hydroxykaempferol-7,6-*O*-diglucoside, tricin *O*-saccharic acid, luteolin 7-*O*-β-D-glucosyl-6-C-α-L-arabinose, chrysoeriol *O*-glucuronic acid, phloretin 2′-*O*-glucoside, cyanidin 3-rutinoside, and cyanin chloride), phenolic acids (9 metabolites, e.g., 3-*O*-*p*-coumaroyl shikimic acid *O*-hexoside, pyrocatechol, syringin, 5-*O*-*p*-coumaroyl quinic acid *O*-hexoside, 1-*O*-[(E)-*p*-cumaroyl]-β-D-glucopyranose, 2,5-dihydroxy benzoic acid *O*-hexside, protocatechuic acid-4-glucoside, 4-methylcatechol, and rosmarinyl glucoside), lignans and coumarins (4 metabolites, e.g., esculin, terpineol monoglucoside, pinoresinol hexose, and matairesinoside), nucleotides and derivatives (1 metabolite, e.g., 2′-deoxyadenosine-5′-monophosphate), and others (1 metabolite, e.g., dihydro-N-caffeoyltyramine). Among them, the relative levels of 18 metabolites decreased more than 5-fold, e.g., acacetin-7-*O*-β-D-glucoside, acacetin-7-*O*-galactoside, tricin 7-*O*-hexoside, chrysoeriol, 3-*O*-*p*-coumaroyl shikimic acid *O*-hexoside, pyrocatechol, syringin, 5-*O*-*p*-coumaroyl quinic acid *O*-hexoside, 1-*O*-[(E)-*p*-cumaroyl]-β-D-glucopyranose, 2,5-dihydroxybenzoic acid *O*-hexoside, protocatechuic acid-4-glucoside, 4-methylcatechol, diosmetin-7-*O*-galactoside, terpineol monoglucoside, pinoresinol-hexose, matairesinoside, quercetin 3,7-bis-*O*-β-D-glucoside, and esculin (Figure 4).

**Figure 4 molecules-28-01722-f004:**
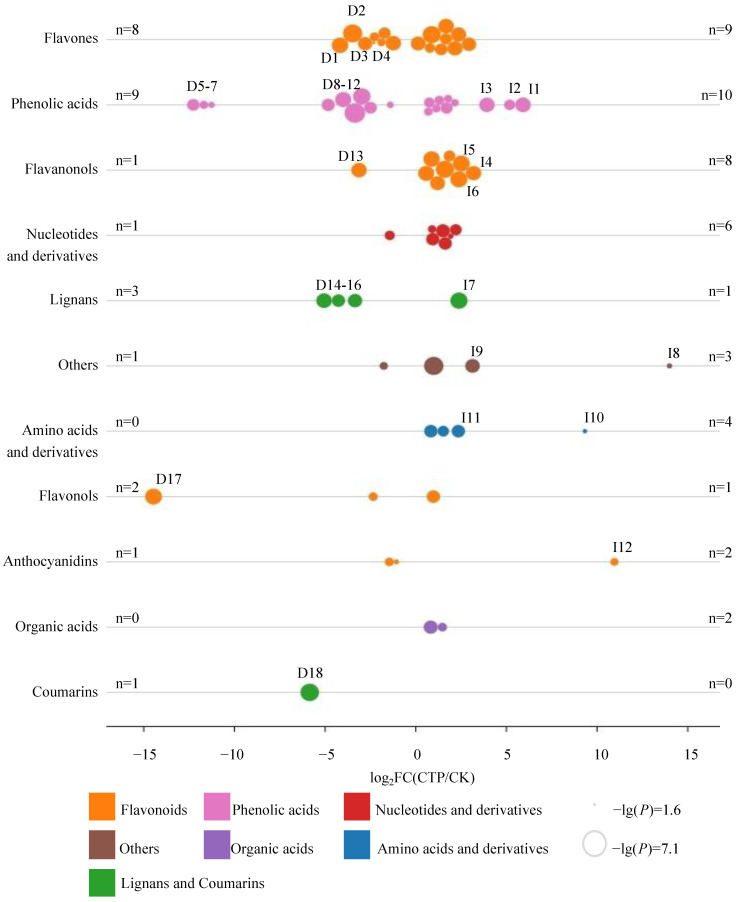
The differentially changed metabolites (DCMs) after catalysis with PPO. D1–18: acacetin-7-*O*-β-D-glucoside, acacetin-7-*O*-galactoside, tricin 7-*O*-hexoside, chrysoeriol, 3-*O*-*p*-coumaroyl shikimic acid *O*-hexoside, pyrocatechol, syringin, 5-*O*-*p*-coumaroyl quinic acid *O*-hexoside, 1-*O*-[(E)-*p*-cumaroyl]-β-D-glucopyranose, 2,5-dihydroxy benzoic acid *O*-hexside, protocatechuic acid-4-glucoside, 4-methylcatechol, diosmetin-7-*O*-galactoside, terpineol monoglucoside, pinoresinol-hexose, matairesinoside, quercetin 3,7-bis-*O*-β-D-glucoside, esculin. I1–12: coniferyl alcohol, esculetin, salicin, theaflavin 3,3′-digallate, theaflavin-3-gallate, theaflavin-3′-gallate, pinoresinol, MAG (18:1) isomer 2, resveratrol, L-methionine, L-homocystine, and peonidin.

Meanwhile, the relative levels of 45 metabolites in CTP/CK were increased significantly (VIP > 1.0, *p* < 0.05, and FC > 2), including flavonoids (19 metabolites, e.g., herbacetin, naringenin chalcone, taxifolin, pinobanksin, 5,7-dihydroxy-3′,4′,5′-trimethoxyflavone, pinocembrin, apigenin, 3′,4′,7-trihydroxyflavone, luteolin, diosmetin, pratensein, theaflavin, jaceosidin, hispidulin, acacetin, theaflavin-3′-gallate, theaflavin-3-gallate, theaflavin 3,3′-digallate, and peonidin), phenolic acids (10 metabolites, e.g., ferulic acid, 4-aminobenzoic acid, vanillin, trans-ferulic acid, caffeic acid, coniferaldehyde, oxalic acid, salicin, esculetin, and coniferyl alcohol), nucleotides and derivatives (6 metabolites, e.g., guanosine 3′,5′-cyclic monophosphate, cytidine, guanosine, 8-hydroxyguanosine, 3′-aenylic acid, and xanthine), amino acids and derivatives (4 metabolites, e.g., leucylphenylalanine, DL-alanyl-DL-phenylalanine, L-homocystine, and L-methionine), organic acids (2 metabolites, e.g., D-glucoronic acid, 5-hydroxyhexanoic acid), lignans and coumarins (1 metabolite, e.g., pinoresinol), and others (3 metabolites, e.g., pyridoxine, resveratrol, and MAG (18:1) isomer 2). Among them, the relative levels of 12 metabolites increased more than 5-fold, including coniferyl alcohol, esculetin, salicin, theaflavin 3,3′-digallate, theaflavin-3-gallate, theaflavin-3′-gallate, pinoresinol, MAG (18:1) isomer 2, resveratrol, L-methionine, L-homocystine, and peonidin (Figure 4). Interestingly, the relative levels of four major theaflavins in black tea, including theaflavin (TF_1_), theaflavin-3-gallate (TF_2_A), theaflavin-3′-gallate (TF_2_B), and theaflavin 3,3′-digallate (TF_3_) [33,34], increased significantly (VIP > 1.0, *p* < 0.05, and FC > 2) in CTP/CK (Figure 4).

In comparison with CK, the levels of EC, EGC, ECG, and EGCG in CTP decreased from 2.94 mg/g, 9.34 mg/g, 17.60 mg/g, 31.98 mg/g to 0.68 mg/g, 2.04 mg/g, 6.05 mg/g, and 8.96 mg/g, respectively, whereas the levels of TF_1_, TF_2_A, TF_2_B, and TF_3_ increased 3.82-, 5.11-, 5.92-, and 6.01-fold, respectively (Figure 5). We suggested that TF_1_ was synthesized through the polymerization of EC and EGC under the catalysis of PPO; PPO could catalyze the polymerization of EC and EGCG to form TF_2_A; ECG and EGC could be polymerized to form TF_2_B under the catalysis of PPO; and TF_3_ was synthesized through the polymerization of ECG and EGCG under the catalysis of PPO. Theaflavins are the general name for a class of compounds with a benzodiazepine structure formed by the condensation of catechins under the catalytic action of PPO [35,36], and they have great potential and broad application prospects in the fields of food, health products, and natural medicine [37,38,39]. In fresh tea leaves, the phenolic hydroxyl groups on the B ring of catechins are oxidized by PPO to form theaflavin intermediates (*o*-quinones) [40,41], which are easily oxidized and polymerized to form theaflavins [42,43]. Therefore, it is proved that TF_1_, TF_2_A, TF_2_B, and TF_3_ can be produced by enzymatic oxidation of PPO only in the presence of dihydroxy-B-cycloflavanol (e.g., EC and ECG) and trihydroxy-B-cycloflavanol (e.g., EGC and EGCG) through the structural formula and the change of the levels of metabolites (Figure 5).

## 3. Materials and Methods

### 3.1. Materials and Chemical Standards

The raw material (RM) was sun-dried green tea leaves with one bud and three leaves, which were collected from Pu’er City Institute of Tea Science, Yunnan Province, China. PPO (500 U/mg) was purchased from Shanghai Yuanye Biotechnology Co., Ltd. (Shanghai, China). Gallic acid (GA), ellagic acid (EA), caffeine (Ca), rutin (Rt), myricetin (My), taxifolin (Ti), quercetin (Qc), kaempferol (Kp), luteolin (Lu), and catechins including (+)-catechin (C), (−)-epicatechin (EC), (−)-epigallocatechin (EGC), (−)-epicatechin 3-*O*-gallate (ECG), (−)-epigallocatechin 3-*O*-gallate (EGCG), (−)-gallocatechin (GC), (−)-gallocatechin gallate (GCG), and (−)-catechin gallate (CG) of high-performance liquid chromatography (HPLC) grade were purchased from Manster Biotechnology Co., Ltd. (Chengdu, China).

### 3.2. Optimization of the PPO Catalysis Conditions

Sun-dried green tea leaves were ground to fine powder with liquid nitrogen 30 min to obtain tea leaves with broken cell walls, and the tea powder can be passed through the 40 mesh sieve. After that, 1 g tea powder was added to 1 mL PPO at a concentration of 0 U/mL (control check, CK), 100 U/mL, 200 U/mL, 300 U/mL, 400 U/mL, and 500 U/mL, respectively, and the reaction was carried out at 35 °C for 7 h, then terminated by boiling water for 10 min. Since then, 1 mL PPO (500 U/mL) was added to the tea powder (1 g) and the reaction was terminated after 6 h, 7 h, and 8 h at 35 °C. TPs were extracted with the methanol extraction method and subjected to HPLC analysis described in our previous report [44]. Briefly, 1 g of sample was extracted with 44.00 mL of methanol:hydrochloric acid (40:4, *v*/*v*) in a flask equipped with a reflux condenser. The extraction was performed in a water bath (at 85 °C) for 90 min. The extractions were diluted to 50 mL, filtered through a 0.2 μm nylon filter, and then analyzed directly by HPLC. Samples were determined using an Agilent 1200 series HPLC system consisting of an LC-20AB solvent delivery unit, an SIL-20A autosampler, a CTO-20A column oven (35 °C), a G1314B UV variable wavelength detector, and an LC Ver1.23 workstation (Agilent Technologies, Santa Clara, CA, USA). Partitioning was performed using an Agilent Poroshell 120 EC-C18 column (4.6 × 100 mm, 2.7 μm) fitted with a C_18_ guard column (Agilent Technologies). The mobile phase was a mixture of (A) 5% acetonitrile and 0.261% ortho-phosphoric acid in water and (B) 80% methanol in water. In an elution gradient, from 0–16 min, buffer B was increased from 10 to 45%; from 16–22 min, buffer B was increased to 65%; and from 22–25.9 min, buffer B was increased to 100%. Three replicates of each sample were extracted, and each extraction was analyzed twice.

### 3.3. Metabolomics Analysis

Metabolites in the tea leaves were extracted and analyzed using a liquid chromatography electrospray ionization tandem mass spectrometry (LC-ESI-MS/MS) based metabolomics approach performed by Metware Biotechnology Co. Ltd., Wuhan, China. The catalyzed tea powder (CTP) or CK samples (100 mg) were weighed and extracted overnight at 4 °C using 1.0 mL 70% methanol. The samples were then centrifuged at 10,000× *g* for 10 min. The supernatant was filtered using a microporous membrane (SCAA-104, 0.22-μm pore size, ANPEL, Shanghai, China) for LC-ESI-MS/MS analysis. Quality control (QC) samples were prepared by mixing sample extracts to examine the repeatability of the analysis.

Samples were injected into an LC-ESI-MS/MS system (UPLC, Shim-pack UFLC Shimadzu CBM30A system, MS, Applied Biosystems 4500 Q-Trap). The LC-ESI-MS/MS system analytical method was performed as described previously [45]. The chromatographic separation was performed on a Waters ACQUITY UPLC HSS T3 C18 column (2.1 × 100 mm, 1.8 μm; Waters Corporation, Milford, MA, USA) at 40 ℃, and the LC parameters were as follows: injection volume, 4μL; flow rate, 0.35 mL/min; The mobile phase was a mixture of (A) 0.04% acetic acid in water and (B) 0.04% acetic acid in acetonitrile; the gradient elution was carried out: 5–95% B for 0–10 min, 95% B for 10–11 min, 5% B for 11–11.1 min, 5% B for 11.1–14 min, and 100% B for 35–45 min. The effluent was alternatively connected to an ESI-triple quadrupole-linear ion trap (QTRAP)-MS controlled by Analyst 1.6.3 software (AB Sciex, Darmstadt, Germany). The operating parameters of the ESI source were as follows: ESI source temperature, 500 ℃; ion spray voltage, 5500 V; ion source gas I (GSI), gas II (GSII), and curtain gas (CUR), 30 psi; and collision-activated dissociation, highest setting. Triple quadrupole (QQQ) scans were acquired as multiple reaction monitoring (MRM) experiments using optimized declustering potentials (DP) and collision energies (CE) for each individual MRM transition. Instrument tuning and mass calibration were performed with 10 μmol/L and 100 μmol/L polypropylene glycol solutions in QQQ and LIT modes, respectively. QQQ scans were acquired as MRM experiments with collision gas (nitrogen) set to 5 psi. DP and CE for individual MRM transitions were carried out with further DP and CE optimization. A specific set of MRM transitions was monitored for each period according to the metabolites eluted within this period [46].

Data filtering, peak detection, alignment, and calculations were performed using Analyst 1.6.3 software (AB Sciex). To facilitate the identification/annotation of metabolites, accurate *m*/*z* ratios were obtained for each precursor ion. Total ion chromatograms and extracted ion chromatograms of QC samples were exported to give an overview of the metabolite profiles of all samples. Metabolites were characterized by searching internal and public databases (MassBank, KNApSAcK, HMDB, MoTo DB, and METLIN) and comparing their *m*/*z* values, retention times, and fragmentation patterns with those of the standards [47,48] and comparing their *m*/*z* values, retention times and fragmentation patterns with those of the standards. The chromatographic peak area of each was calculated. Positive and negative data were combined to obtain a combined data set.

### 3.4. Statistical Analysis

Statistical analyses were performed using IBM SPSS Statistics 26.0 (SPSS Inc., Chicago, IL, USA). Principal component analysis (PCA) and orthogonal partial least square discriminant analysis (OPLS-DA) results were generated by SIMCA 14.1 (Umetrics, Umea, Sweden) to visualize the metabolic differences between the experimental groups after normalization and standardization processing. Variable importance in projection (VIP) analysis ranked the overall contribution of each variable to the OPLS-DA model, and those variables with VIP > 1.0, *p* < 0.05, and fold change (FC) > 2 or < 0.5 were classified as differentially changed metabolites (DCMs) [49].

## 4. Conclusions

The PPO catalytic conditions on TPs in liquid nitrogen grinding sun-dried green tea leaves were optimized, and 500 U/mL PPO reaction for 7 h can catalyze TPs effectively. Meanwhile, a total of 441 metabolites were identified in tea leaves, which were classified into 11 classes, including flavonoids (125 metabolites), phenolic acids (67 metabolites), and lipids (55 metabolites), amino acids and derivatives (51 metabolites), nucleotides and derivatives (34 metabolites), organic acids (25 metabolites), alkaloids (22 metabolites), tannins (12 metabolites), lignans and coumarins (11 metabolites), terpenoids (1 metabolite), and others (38 metabolites). Furthermore, the relative levels of 28 metabolites, including flavonoids (13 metabolites), phenolic acids (9 metabolites), lignans and coumarins (4 metabolites), nucleotides and derivatives (1 metabolites), and others (1 metabolite) were decreased significantly after catalysis (VIP > 1.0, *p* < 0.05, and FC < 0.5); the relative levels of 45 metabolites including flavonoids (19 metabolites), phenolic acids (10 metabolites), nucleotides and derivatives (6 metabolites), amino acids and derivatives (4 metabolites), organic acids (2 metabolites), lignans and coumarins (1 metabolite), and others (3 metabolites) were increased significantly (VIP > 1.0, *p* < 0.05, and FC > 2), while, these four major theaflavins (TF_1_, TF_2_A, TF_2_B, and TF_3_) can be produced by enzymatic oxidation of PPO only in the presence of dihydroxy-B-cycloflavanol (e.g., EC and ECG) and trihydroxy-B-cycloflavanol (e.g., EGC and EGCG).

Therefore, an in vitro catalysis of TPs by PPO was established and provided technical references for the study of the catalytic mechanism of PPO in tea leaves.

## Figures and Tables

**Figure 2 molecules-28-01722-f002:**
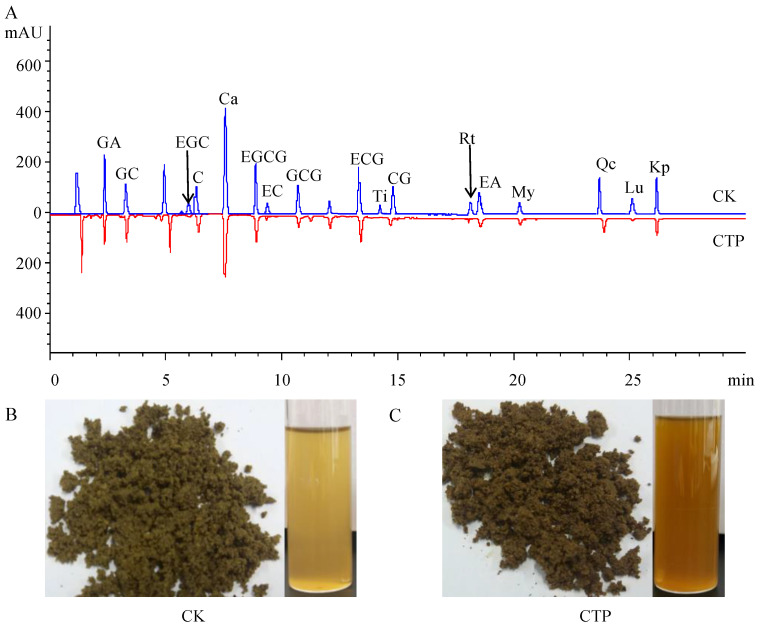
HPLC chromatograms for the determination of TPs in sun-dried green tea leaves (**A**), appearance of tea leaves and infusion in control check (**B**) and catalysis (**C**).

**Figure 3 molecules-28-01722-f003:**
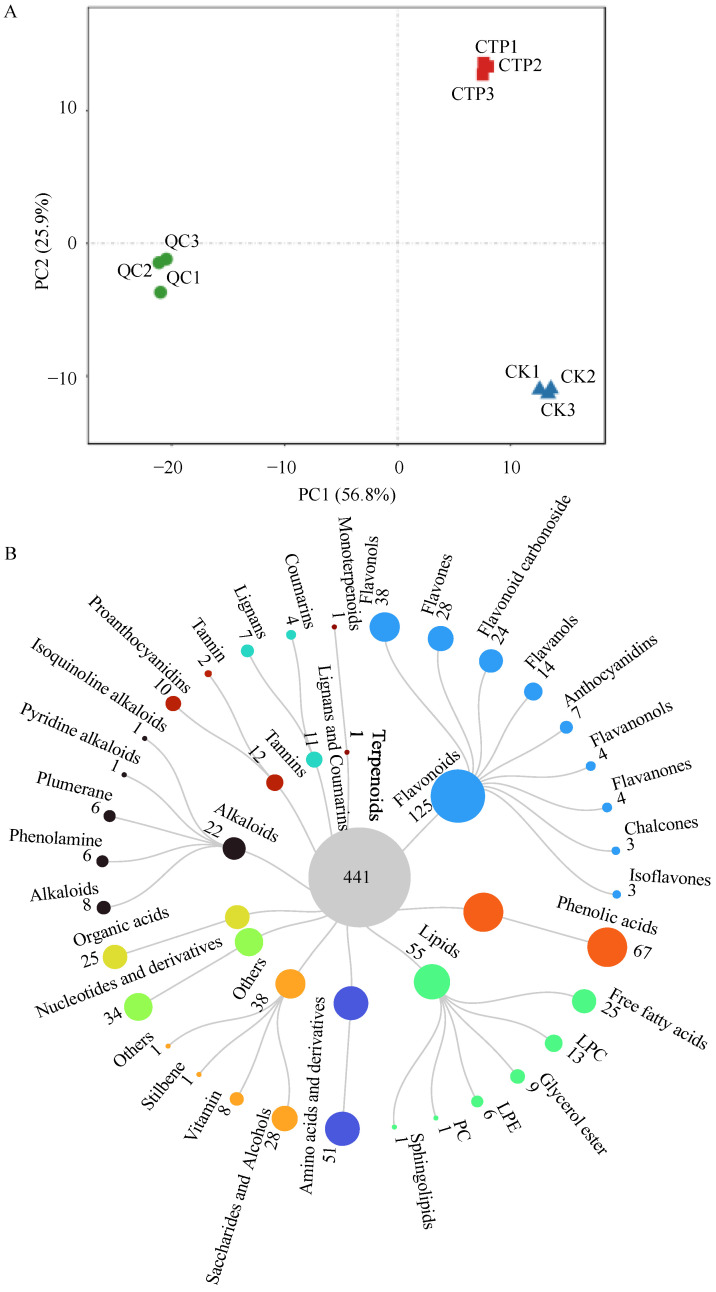
Results of the metabolomics analysis. PCA (**A**) and classification (**B**) of identified metabolites in CTP, CK, and QC. LPC: lysophosphatidylcholine, LPE: lysophosphatidylethanolamine, PC: phosphatidyl cholines.

**Figure 5 molecules-28-01722-f005:**
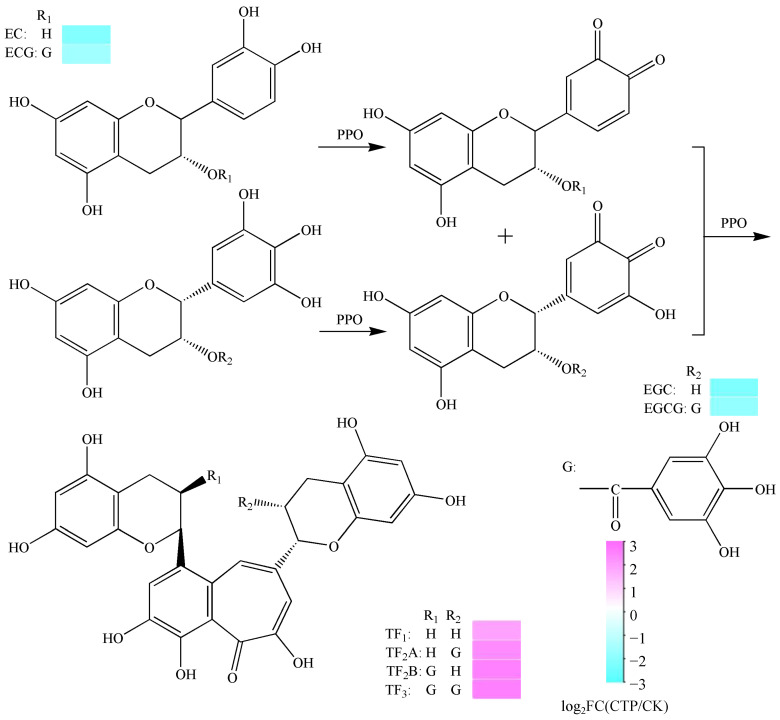
The change of levels and possible formation mechanism of theaflavin (TF_1_), theaflavin-3-gallate (TF_2_A), theaflavin-3′-gallate (TF_2_B), and theaflavin 3,3′-digallate (TF_3_) from catechins catalyzed by PPO.

**Table 1 molecules-28-01722-t001:** The relative levels of metabolites in CTP and CK.

Metabolites	Class	Sub-Class	Relative Level	
			CTP	CK
Pipecolic acid	AAD 1	AAD	0.06%	0.08%
1,2-N-Methylpipecolic acid	AAD	AAD	0.02%	0.02%
L-Asparagine anhydrous	AAD	AAD	0.00%	0.00%
L-Homocitrulline	AAD	AAD	0.00%	0.00%
Trans-4-Hydroxy-L-proline	AAD	AAD	0.41%	0.30%
L-Aspartic acid	AAD	AAD	0.41%	0.41%
L-Leucine	AAD	AAD	0.14%	0.14%
L-(−)-Threonine	AAD	AAD	0.00%	0.00%
L-(−)-Tyrosine	AAD	AAD	0.34%	0.44%
L-Histidine	AAD	AAD	0.01%	0.01%
L-Valine	AAD	AAD	4.54%	4.50%
L-Isoleucine	AAD	AAD	0.12%	0.12%
L-(+)-Arginine	AAD	AAD	0.02%	0.02%
L-Pyroglutamic acid	AAD	AAD	0.01%	0.01%
N-Acetyl-L-tyrosine	AAD	AAD	0.01%	0.00%
Phe-phe	AAD	AAD	0.00%	0.00%
N-Glycyl-L-leucine	AAD	AAD	0.05%	0.03%
5-Oxoproline	AAD	AAD	0.01%	0.00%
D-Serine	AAD	AAD	0.00%	0.00%
L-theanine	AAD	AAD	0.09%	0.10%
Cis-4-Hydroxy-D-proline	AAD	AAD	0.01%	0.01%
α-Aminocaproic acid	AAD	AAD	0.14%	0.14%
Oxidized glutathione	AAD	AAD	0.01%	0.02%
DL-Alanyl-DL-phenylalanine	AAD	AAD	0.06%	0.03%
Leucylphenylalanine	AAD	AAD	0.10%	0.05%
Glycylisoleucine	AAD	AAD	0.06%	0.03%
Glycylphenylalanine	AAD	AAD	0.05%	0.05%
Acetyltryptophan	AAD	AAD	0.20%	0.19%
L-Proline	AAD	AAD	0.27%	0.27%
L-Citrulline	AAD	AAD	0.02%	0.03%
L-Glutamic acid	AAD	AAD	0.02%	0.01%
L-(+)-Lysine	AAD	AAD	0.44%	0.33%
N6-Acetyl-L-lysine	AAD	AAD	0.01%	0.01%
N-α-Acetyl-L-glutamine	AAD	AAD	0.01%	0.01%
L-Glutamine	AAD	AAD	0.51%	0.40%
N-Acetyl-L-leucine	AAD	AAD	0.03%	0.02%
L-Tyramine	AAD	AAD	1.28%	1.37%
L-Methionine	AAD	AAD	0.00%	0.00%
N-Acetylaspartate	AAD	AAD	0.00%	0.00%
(5-L-Glutamyl)-L-amino acid	AAD	AAD	0.00%	0.00%
Methionine sulfoxide	AAD	AAD	0.09%	0.08%
4-Hydroxy-L-glutamic acid	AAD	AAD	0.01%	0.01%
L-Homocystine	AAD	AAD	0.04%	0.01%
2-Aminoisobutyric acid	AAD	AAD	0.15%	0.16%
N,N-Dimethylglycine	AAD	AAD	0.02%	0.02%
N-Acetylthreonine	AAD	AAD	0.00%	0.00%
H-HomoArg-OH	AAD	AAD	0.01%	0.01%
N-Acetyl-DL-tryptophan	AAD	AAD	0.04%	0.04%
Tryptophan	AAD	AAD	1.61%	1.81%
Phenylalanine	AAD	AAD	0.11%	0.12%
Proline βine	AAD	AAD	0.01%	0.01%
Spermine	Alkaloids	Alkaloids	0.81%	0.75%
Betaine	Alkaloids	Alkaloids	0.04%	0.05%
Theophylline	Alkaloids	Alkaloids	0.03%	0.03%
*O*-Phosphorylethanolamine	Alkaloids	Alkaloids	0.01%	0.01%
N-Oleoylethanolamine	Alkaloids	Alkaloids	0.02%	0.02%
Caffeine	Alkaloids	Alkaloids	0.28%	0.32%
Choline	Alkaloids	Alkaloids	1.78%	1.87%
Acetylcholine	Alkaloids	Alkaloids	0.00%	0.00%
Isoquinoline	Alkaloids	Isoquinoline alkaloids	0.00%	0.00%
Fer-agmatine	Alkaloids	Phenolamine	0.21%	0.32%
N-p-Coumaroyl putrescine	Alkaloids	Phenolamine	0.01%	0.01%
N-Feruloyl agmatine	Alkaloids	Phenolamine	0.39%	0.65%
Dihydro-N-caffeoyltyramine	Alkaloids	Phenolamine	0.00%	0.00%
N-cis-sinapoyltyramine	Alkaloids	Phenolamine	0.00%	0.00%
N-Trans-feruloyltyramine	Alkaloids	Phenolamine	0.00%	0.00%
Tryptamine	Alkaloids	Plumerane	0.03%	0.03%
Indole-5-carboxylic acid	Alkaloids	Plumerane	0.00%	0.00%
Indole-3-carboxaldehyde	Alkaloids	Plumerane	0.04%	0.03%
N-Acetyl-5-hydroxytryptamine	Alkaloids	Plumerane	0.01%	0.01%
Indole-3-carboxylic acid	Alkaloids	Plumerane	0.01%	0.01%
Indole	Alkaloids	Plumerane	0.01%	0.01%
Trigonelline	Alkaloids	Pyridine alkaloids	0.00%	0.00%
Cyanidin 3-rutinoside	Flavonoids	Anthocyanidins	0.02%	0.04%
Cyanin chloride	Flavonoids	Anthocyanidins	0.05%	0.14%
Cyanidin 3-*O*-galactoside	Flavonoids	Anthocyanidins	0.03%	0.05%
Peonidin	Flavonoids	Anthocyanidins	0.00%	0.00%
Cyanidin *O*-diacetyl-hexoside-*O*-glyceric acid	Flavonoids	Anthocyanidins	0.00%	0.00%
Malvidin 3-*O*-galactoside	Flavonoids	Anthocyanidins	0.00%	0.00%
Malvidin 3-*O*-glucoside	Flavonoids	Anthocyanidins	0.00%	0.00%
Naringenin chalcone	Flavonoids	Chalcones	0.38%	0.19%
Phloretin 2′-*O*-glucoside	Flavonoids	Chalcones	0.00%	0.01%
Phloretin	Flavonoids	Chalcones	0.01%	0.01%
Epigallocatechin gallate	Flavonoids	Flavanols	0.37%	0.43%
(−)-Epigallocatechin	Flavonoids	Flavanols	3.07%	3.92%
(+)-Gallocatechin	Flavonoids	Flavanols	0.11%	0.16%
Catechin	Flavonoids	Flavanols	0.05%	0.06%
Gallate catechin gallate	Flavonoids	Flavanols	0.32%	0.38%
(−)-Epicatechin gallate	Flavonoids	Flavanols	0.12%	0.12%
(−)-Epiafzelechin	Flavonoids	Flavanols	0.26%	0.31%
Gallocatechin 3-*O*-gallate	Flavonoids	Flavanols	0.35%	0.39%
Catechin gallate	Flavonoids	Flavanols	0.08%	0.09%
Catechin-catechin-catechin	Flavonoids	Flavanols	0.01%	0.02%
Epicatechin-epiafzelechin	Flavonoids	Flavanols	0.01%	0.01%
L-Epicatechin	Flavonoids	Flavanols	0.18%	0.22%
Catechin-(7,8-bc)-4β-(3,4-dihydroxyphenyl)-dihydro-2-(3H)-pyranone	Flavonoids	Flavanols	0.01%	0.01%
Catechin-(7,8-bc)-4α-(3,4-dihydroxyphenyl)-dihydro-2-(3H)-pyranone	Flavonoids	Flavanols	0.09%	0.13%
Diosmetin	Flavonoids	Flavanones	0.07%	0.02%
Butin	Flavonoids	Flavanones	0.41%	0.24%
Diosmetin-6-C-glucoside	Flavonoids	Flavanones	0.00%	0.00%
Diosmetin-7-*O*-galactoside	Flavonoids	Flavanones	0.01%	0.08%
Taxifolin	Flavonoids	Flavanonols	0.04%	0.02%
Dihydromyricetin	Flavonoids	Flavanonols	0.04%	0.04%
Pinobanksin	Flavonoids	Flavanonols	0.32%	0.15%
Pinocembrin	Flavonoids	Flavanonols	0.01%	0.00%
Acacetin	Flavonoids	Flavones	0.01%	0.00%
Apigenin 5-*O*-glucoside	Flavonoids	Flavones	0.02%	0.02%
Tricetin	Flavonoids	Flavones	0.19%	0.20%
5,7-Dihydroxy-3′,4′,5′-trimethoxyflavone	Flavonoids	Flavones	0.01%	0.00%
Luteolin *O*-hexosyl-*O*-pentoside	Flavonoids	Flavones	0.00%	0.00%
Luteolin 3′,7-di-*O*-glucoside	Flavonoids	Flavones	0.00%	0.00%
Tricin 7-*O*-hexoside	Flavonoids	Flavones	0.04%	0.28%
Acacetin-*O*-glucuronic acid	Flavonoids	Flavones	0.03%	0.03%
Apigenin 7-*O*-glucoside	Flavonoids	Flavones	0.00%	0.00%
Chrysoeriol *O*-glucuronic acid	Flavonoids	Flavones	0.00%	0.00%
Tricin *O*-saccharic acid	Flavonoids	Flavones	0.00%	0.01%
Luteolin-7-*O*-glucoside	Flavonoids	Flavones	0.10%	0.10%
Luteolin-7-*O*-β-D-glucuronide	Flavonoids	Flavones	0.00%	0.00%
Luteolin-7-*O*-β-D-rutinoside	Flavonoids	Flavones	0.06%	0.06%
Hispidulin	Flavonoids	Flavones	0.18%	0.04%
Ladanein	Flavonoids	Flavones	0.00%	0.00%
Jaceosidin	Flavonoids	Flavones	0.26%	0.06%
5-Hydroxy-6,7,3′,4′-tetramethoxyflavone	Flavonoids	Flavones	1.09%	1.49%
Luteolin 7-*O*-β-D-glucosyl-6-C-α-L-arabinose	Flavonoids	Flavones	0.00%	0.00%
3′,4′,7-Trihydroxyflavone	Flavonoids	Flavones	0.00%	0.00%
Apigenin	Flavonoids	Flavones	0.03%	0.01%
Luteolin	Flavonoids	Flavones	0.05%	0.02%
Acacetin-7-*O*-galactoside	Flavonoids	Flavones	0.00%	0.04%
Tilianin	Flavonoids	Flavones	0.00%	0.01%
Luteolin-7-*O*-rutinoside	Flavonoids	Flavones	0.14%	0.15%
Luteolin-7,3′-Di-*O*-β-D-glucoside	Flavonoids	Flavones	0.05%	0.03%
Lonicerin	Flavonoids	Flavones	0.39%	0.46%
Chrysoeriol	Flavonoids	Flavones	0.00%	0.00%
Vitexin	Flavonoids	Flavonoid carbonoside	0.08%	0.08%
Apigenin 6,8-C-diglucoside	Flavonoids	Flavonoid carbonoside	0.41%	0.43%
Isoschaftoside	Flavonoids	Flavonoid carbonoside	1.19%	1.33%
Orientin	Flavonoids	Flavonoid carbonoside	0.27%	0.31%
Isovitexin	Flavonoids	Flavonoid carbonoside	0.08%	0.08%
Schaftoside	Flavonoids	Flavonoid carbonoside	0.01%	0.01%
C-Hexosyl-luteolin C-pentoside	Flavonoids	Flavonoid carbonoside	0.00%	0.00%
6-C-Hexosyl luteolin *O*-pentoside	Flavonoids	Flavonoid carbonoside	0.00%	0.00%
C-Hexosyl-apigenin *O*-pentoside	Flavonoids	Flavonoid carbonoside	0.02%	0.02%
Di-C,C-hexosyl-apigenin	Flavonoids	Flavonoid carbonoside	0.62%	0.66%
C-Hexosyl-luteolin *O*-*p*-coumaroylhexoside	Flavonoids	Flavonoid carbonoside	0.00%	0.01%
Luteolin 8-C-hexosyl-*O*-hexoside	Flavonoids	Flavonoid carbonoside	0.27%	0.30%
C-Hexosyl-apigenin *O*-*p*-coumaroylhexoside	Flavonoids	Flavonoid carbonoside	0.02%	0.02%
Apigenin 8-C-pentoside	Flavonoids	Flavonoid carbonoside	0.22%	0.28%
Chrysoeriol C-hexoside	Flavonoids	Flavonoid carbonoside	0.00%	0.01%
Luteolin C-hexoside	Flavonoids	Flavonoid carbonoside	0.02%	0.02%
Isohemiphloin	Flavonoids	Flavonoid carbonoside	0.04%	0.05%
Isovitexin 7-*O*-glucoside	Flavonoids	Flavonoid carbonoside	0.00%	0.00%
Vitexin 2″-*O*-β-L-rhamnoside	Flavonoids	Flavonoid carbonoside	0.16%	0.18%
Luteolin-6,8-di-C-glucoside	Flavonoids	Flavonoid carbonoside	0.01%	0.01%
Apigenin-6-C-2-glucuronylxyloside	Flavonoids	Flavonoid carbonoside	0.07%	0.08%
Isoorientin	Flavonoids	Flavonoid carbonoside	0.16%	0.17%
Vitexin-2-*O*-D-glucopyranoside	Flavonoids	Flavonoid carbonoside	0.07%	0.08%
Apigenin-6-C-β-D-xyloside-8-C-β-darabinoside	Flavonoids	Flavonoid carbonoside	1.07%	1.27%
Isorhamnetin-3-*O*-rutinoside	Flavonoids	Flavonols	0.01%	0.01%
Kaempferol-3-*O*-glucoside-7-*O*-rhamnoside	Flavonoids	Flavonols	0.43%	0.50%
Quercetin-3-*O*-glucoside-7-*O*-rhamnoside	Flavonoids	Flavonols	0.27%	0.30%
Quercetin 3-*O*-rhanosylgalactoside	Flavonoids	Flavonols	0.27%	0.32%
Myricetin	Flavonoids	Flavonols	0.05%	0.04%
Quercitrin	Flavonoids	Flavonols	0.36%	0.37%
Myricitrin	Flavonoids	Flavonols	0.01%	0.01%
Rutin	Flavonoids	Flavonols	1.47%	1.55%
Hyperin	Flavonoids	Flavonols	0.37%	0.36%
Isorhamnetin	Flavonoids	Flavonols	0.00%	0.00%
Kaempferol 7-*O*-glucosdie	Flavonoids	Flavonols	0.58%	0.70%
Spiraeoside	Flavonoids	Flavonols	0.17%	0.17%
Trifolin	Flavonoids	Flavonols	0.45%	0.52%
Kaempferin	Flavonoids	Flavonols	0.02%	0.02%
Kaempferol	Flavonoids	Flavonols	0.15%	0.07%
Tiliroside	Flavonoids	Flavonols	0.23%	0.31%
Herbacetin	Flavonoids	Flavonols	0.00%	0.00%
Gossypitrin	Flavonoids	Flavonols	0.18%	0.15%
Avicularin	Flavonoids	Flavonols	1.02%	1.17%
Astragalin	Flavonoids	Flavonols	0.62%	0.74%
Quercetin-3-*O*-α-L-arabinopyranoside	Flavonoids	Flavonols	0.17%	0.21%
Quercetin *O*-acetylhexoside	Flavonoids	Flavonols	0.00%	0.01%
Di-*O*-methylquercetin	Flavonoids	Flavonols	0.04%	0.04%
Kaempferol 7-*O*-rhamnoside	Flavonoids	Flavonols	0.02%	0.02%
Kaempferol 3-*O*-rutinoside	Flavonoids	Flavonols	0.95%	0.88%
Kaempferol 3,7-dirhamnoside	Flavonoids	Flavonols	0.01%	0.01%
Quercetin	Flavonoids	Flavonols	0.26%	0.15%
Quercetin 3-*O*-glucoside	Flavonoids	Flavonols	0.21%	0.13%
Bioquercetin	Flavonoids	Flavonols	0.03%	0.04%
Juglanin	Flavonoids	Flavonols	0.00%	0.01%
3,5,6,7,8,3′,4′-Heptamethoxyflavone	Flavonoids	Flavonols	0.06%	0.06%
Isoquercitrin	Flavonoids	Flavonols	1.01%	1.11%
Quercetin-7-*O*-(6′-*O*-malonyl)-β-D-glucoside	Flavonoids	Flavonols	0.09%	0.10%
Quercetin 3,7-bis-*O*-β-D-glucoside	Flavonoids	Flavonols	0.00%	0.01%
6-Hydroxykaempferol-7-*O*-glucoside	Flavonoids	Flavonols	0.77%	0.85%
6-Hydroxykaempferol-3,6-*O*-diglucoside	Flavonoids	Flavonols	0.02%	0.03%
6-Hydroxykaempferol-7,6-*O*-diglucoside	Flavonoids	Flavonols	0.00%	0.01%
6-Hydroxykaempferol-3-*O*-rutin-6-*O*-glucoside	Flavonoids	Flavonols	0.00%	0.00%
Genistein 8-C-apiosyl(1→6)glucoside	Flavonoids	Isoflavones	0.19%	0.21%
Genistein 8-C-glucoside	Flavonoids	Isoflavones	0.86%	0.95%
Pratensein	Flavonoids	Isoflavones	0.15%	0.04%
Esculin	LC 2	Coumarins	0.00%	0.04%
7-Methoxycoumarin	LC	Coumarins	0.13%	0.16%
1-Methoxyphaseollin	LC	Coumarins	0.01%	0.01%
Fraxetin	LC	Coumarins	0.04%	0.03%
Pinoresinol-hexose	LC	Lignans	0.00%	0.10%
Pinoresinol	LC	Lignans	1.03%	0.19%
Terpineol monoglucoside	LC	Lignans	0.00%	0.10%
Medioresinol	LC	Lignans	0.01%	0.01%
Syringaresinol	LC	Lignans	0.00%	0.00%
Matairesinoside	LC	Lignans	0.00%	0.02%
Citropten	LC	Lignans	0.00%	0.01%
13-Oxo-9-hydroxy-10-octadecenoic acid	Lipids	Free fatty acids	0.01%	0.01%
9,10-Dihydroxy-12-octadecenoic acid	Lipids	Free fatty acids	0.02%	0.02%
13-Hydroxy-9,11-octadecadienoic acid	Lipids	Free fatty acids	0.30%	0.28%
9-Hydroxy-10,12-octadecadienoic acid	Lipids	Free fatty acids	0.30%	0.28%
Octadecenoic amide	Lipids	Free fatty acids	0.01%	0.02%
Myristic acid	Lipids	Free fatty acids	1.71%	1.70%
Pentadecanoic acid	Lipids	Free fatty acids	0.01%	0.01%
Palmitoleic acid	Lipids	Free fatty acids	0.00%	0.00%
γ-Linolenic acid	Lipids	Free fatty acids	1.73%	1.62%
Cis-10-Heptadecenoic acid	Lipids	Free fatty acids	0.22%	0.23%
Elaidic acid	Lipids	Free fatty acids	2.79%	2.63%
Dodecanedioic acid	Lipids	Free fatty acids	0.00%	0.00%
Undecylic acid	Lipids	Free fatty acids	0.04%	0.04%
Stearic acid	Lipids	Free fatty acids	3.45%	3.32%
Linoleic acid	Lipids	Free fatty acids	0.01%	0.01%
11-Octadecanoic acid	Lipids	Free fatty acids	0.88%	0.82%
Punicic acid	Lipids	Free fatty acids	1.09%	1.05%
9,10-EODE	Lipids	Free fatty acids	0.52%	0.47%
9-HOTrE	Lipids	Free fatty acids	0.28%	0.25%
Hexadecanoic acid 2,3-dihydroxypropyl ester	Lipids	Free fatty acids	0.05%	0.05%
Eicosadienoic acid	Lipids	Free fatty acids	0.03%	0.03%
10,16-Dihydroxy-palmitic acid	Lipids	Free fatty acids	0.01%	0.01%
9-Hydroxy-12-oxo-10-octadecenoic acid	Lipids	Free fatty acids	0.04%	0.03%
9,12,13-Trihyroxy-10,15-octadecadienoic acid	Lipids	Free fatty acids	0.02%	0.02%
9,10,13-Trihyroxy-11-octadecadienoic acid	Lipids	Free fatty acids	0.05%	0.05%
MAG (18:4) isomer 1	Lipids	Glycerol ester	0.07%	0.12%
MAG (18:2) isomer 1	Lipids	Glycerol ester	0.01%	0.02%
MAG (18:1) isomer 2	Lipids	Glycerol ester	0.01%	0.00%
MAG (18:2)	Lipids	Glycerol ester	0.01%	0.01%
MAG (18:3) isomer 3	Lipids	Glycerol ester	1.53%	1.57%
MAG (18:3) isomer 4	Lipids	Glycerol ester	0.00%	0.00%
MAG (18:1) isomer 1	Lipids	Glycerol ester	0.01%	0.01%
MAG (18:3) isomer 1	Lipids	Glycerol ester	0.19%	0.17%
Glyceryl linoleate	Lipids	Glycerol ester	0.03%	0.03%
1-Stearoyl-sn-glycero-3-phosphocholine	Lipids	LPC 3	0.85%	1.01%
LysoPC 18:3	Lipids	LPC	0.18%	0.21%
LysoPC 16:0	Lipids	LPC	0.04%	0.05%
LysoPC 16:2 (2n isomer)	Lipids	LPC	0.03%	0.03%
LysoPC 15:0	Lipids	LPC	0.12%	0.14%
LysoPC 14:0 (2n isomer)	Lipids	LPC	0.04%	0.04%
LysoPC 16:0 (2n isomer)	Lipids	LPC	0.05%	0.05%
LysoPC 18:0	Lipids	LPC	0.29%	0.35%
PC (18:2) isomer	Lipids	LPC	0.13%	0.14%
LysoPC (16:1)	Lipids	LPC	0.89%	1.05%
LysoPC (18:2)	Lipids	LPC	0.13%	0.14%
LysoPC (18:1)	Lipids	LPC	0.02%	0.02%
LysoPC (18:0)	Lipids	LPC	0.81%	0.96%
LysoPE 18:1	Lipids	LPE 4	0.20%	0.24%
LysoPE 18:1 (2n isomer)	Lipids	LPE	0.12%	0.14%
LysoPE 14:0	Lipids	LPE	0.01%	0.01%
LysoPE 18:2 (2n isomer)	Lipids	LPE	0.75%	0.83%
LysoPE 16:0	Lipids	LPE	1.42%	1.64%
LysoPE 16:0 (2n isomer)	Lipids	LPE	0.40%	0.38%
PC (18:2)	Lipids	PC 5	0.13%	0.14%
Hexadecylsphingosine	Lipids	Sphingolipids	0.49%	0.57%
Uridine	ND 6	ND	0.12%	0.11%
Thymine	ND	ND	0.01%	0.01%
Cytosine	ND	ND	0.05%	0.03%
5-Methylcytosine	ND	ND	0.02%	0.02%
Guanosine 3′,5′-cyclic monophosphate	ND	ND	1.93%	0.86%
Xanthosine	ND	ND	0.05%	0.04%
8-Hydroxyguanosine	ND	ND	0.00%	0.00%
1-Methyladenine	ND	ND	0.01%	0.01%
3′-Aenylic acid	ND	ND	0.02%	0.01%
β-Pseudouridine	ND	ND	0.02%	0.01%
Nicotinic acid adenine dinucleotide	ND	ND	0.00%	0.00%
Adenosine 5′-monophosphate	ND	ND	0.00%	0.00%
Hypoxanthine	ND	ND	0.01%	0.01%
Adenine	ND	ND	0.01%	0.01%
2-Hydroxy-6-aminopurine	ND	ND	0.01%	0.01%
Adenosine	ND	ND	1.00%	0.54%
Xanthine	ND	ND	0.01%	0.00%
Uracil	ND	ND	0.00%	0.00%
Thymidine	ND	ND	0.06%	0.06%
Guanine	ND	ND	0.07%	0.10%
Allopurinol	ND	ND	0.00%	0.00%
Guanosine	ND	ND	1.28%	0.42%
Deoxyguanosine	ND	ND	0.09%	0.10%
Deoxycytidine	ND	ND	0.01%	0.01%
3-Methylxanthine	ND	ND	0.76%	1.09%
2′-Deoxycytidine-5′-monophosphate	ND	ND	0.00%	0.00%
5′-Deoxy-5′-(methylthio)adenosine	ND	ND	0.76%	0.71%
7-Methylxanthine	ND	ND	0.18%	0.26%
2′-Deoxyadenosine-5′-monophosphate	ND	ND	0.00%	0.00%
N6-Succinyl adenosine	ND	ND	0.06%	0.06%
Cytidine	ND	ND	0.34%	0.14%
Deoxyadenosine	ND	ND	0.19%	0.17%
2-(Dimethylamino)guanosine	ND	ND	0.31%	0.35%
7-Methylguanine	ND	ND	0.02%	0.03%
3-Hydroxy-3-methyl butyric acid	Organic acids	Organic acids	0.06%	0.05%
Shikimic acid	Organic acids	Organic acids	0.10%	0.11%
2-Furanoic acid	Organic acids	Organic acids	0.02%	0.02%
Succinic acid	Organic acids	Organic acids	4.26%	4.75%
Adipic acid	Organic acids	Organic acids	0.03%	0.03%
Anchoic acid	Organic acids	Organic acids	0.21%	0.21%
Kinic acid	Organic acids	Organic acids	1.22%	0.85%
Citric acid	Organic acids	Organic acids	1.03%	1.53%
DL-P-hydroxyphenyllactic acid	Organic acids	Organic acids	0.01%	0.01%
Pipecolinic acid	Organic acids	Organic acids	0.01%	0.01%
Fumaric acid	Organic acids	Organic acids	0.01%	0.01%
Citraconic acid	Organic acids	Organic acids	0.05%	0.05%
Methylmalonic acid	Organic acids	Organic acids	0.36%	0.45%
2-Methylsuccinic acid	Organic acids	Organic acids	0.12%	0.12%
4-Guanidinobutyric acid	Organic acids	Organic acids	0.11%	0.10%
3-Hydroxybutyrate	Organic acids	Organic acids	0.06%	0.07%
Sodium valproate	Organic acids	Organic acids	0.07%	0.07%
2-Methylglutaric acid	Organic acids	Organic acids	0.03%	0.03%
1,3,7-Trimethyluric acid	Organic acids	Organic acids	0.00%	0.00%
5-Hydroxyhexanoic acid	Organic acids	Organic acids	0.01%	0.00%
Aldehydo-D-galacturonate	Organic acids	Organic acids	0.02%	0.01%
Malic acid	Organic acids	Organic acids	0.07%	0.10%
6-Aminocaproic acid	Organic acids	Organic acids	0.56%	0.63%
4-Acetamidobutyric acid	Organic acids	Organic acids	0.13%	0.13%
γ-Aminobutyric acid	Organic acids	Organic acids	0.01%	0.02%
p-Coumaroylferuloyltartaric acid	Phenolic acids	Phenolic acids	0.00%	0.00%
3,4-Dicaffeoylquinic acid	Phenolic acids	Phenolic acids	0.20%	0.19%
Coniferaldehyde	Phenolic acids	Phenolic acids	0.00%	0.00%
Syringin	Phenolic acids	Phenolic acids	0.00%	0.00%
Ferulic acid	Phenolic acids	Phenolic acids	0.07%	0.03%
Gallic acid	Phenolic acids	Phenolic acids	0.37%	0.27%
Coniferyl alcohol	Phenolic acids	Phenolic acids	0.19%	0.00%
Chlorogenic acid methyl ester	Phenolic acids	Phenolic acids	0.01%	0.01%
p-Hydroxyphenyl acetic acid	Phenolic acids	Phenolic acids	0.00%	0.00%
Protocatechuic acid	Phenolic acids	Phenolic acids	1.16%	0.82%
3-Aminosalicylic acid	Phenolic acids	Phenolic acids	0.01%	0.01%
Vanillin	Phenolic acids	Phenolic acids	0.04%	0.02%
3-(4-Hydroxyphenyl)-propionic acid	Phenolic acids	Phenolic acids	0.01%	0.01%
4-Methylcatechol	Phenolic acids	Phenolic acids	0.00%	0.00%
4-Hydroxybenzaldehyde	Phenolic acids	Phenolic acids	0.06%	0.05%
2,3-Dihydroxybenzoic acid	Phenolic acids	Phenolic acids	0.94%	0.69%
4-Hydroxybenzoic acid	Phenolic acids	Phenolic acids	0.05%	0.04%
Anthranilic acid	Phenolic acids	Phenolic acids	0.06%	0.04%
Methyl p-coumarate	Phenolic acids	Phenolic acids	0.02%	0.03%
Trans-4-Hydroxycinnamic acid methyl ester	Phenolic acids	Phenolic acids	0.02%	0.02%
4-Aminobenzoic acid	Phenolic acids	Phenolic acids	0.00%	0.00%
Syringic aldehyde	Phenolic acids	Phenolic acids	0.00%	0.00%
Trans-ferulic acid	Phenolic acids	Phenolic acids	0.06%	0.02%
Pyrocatechol	Phenolic acids	Phenolic acids	0.00%	0.00%
Salicin	Phenolic acids	Phenolic acids	0.00%	0.00%
Cryptochlorogenic acid	Phenolic acids	Phenolic acids	1.52%	1.48%
Caffeic acid	Phenolic acids	Phenolic acids	0.08%	0.03%
Cinnamic acid	Phenolic acids	Phenolic acids	0.00%	0.00%
Tyrosol	Phenolic acids	Phenolic acids	0.00%	0.00%
Hydroxy-methoxycinnamate	Phenolic acids	Phenolic acids	0.00%	0.00%
3-*O*-Feruloyl quinic acid	Phenolic acids	Phenolic acids	0.11%	0.12%
2,5-Dihydroxy benzoic acid *O*-hexside	Phenolic acids	Phenolic acids	0.01%	0.11%
5-*O*-*p*-Coumaroyl quinic acid *O*-hexoside	Phenolic acids	Phenolic acids	0.00%	0.02%
1-*O*-*p*-Coumaroyl quinic acid	Phenolic acids	Phenolic acids	0.30%	0.34%
3-*O*-*p*-coumaroyl shikimic acid *O*-hexoside	Phenolic acids	Phenolic acids	0.00%	0.00%
Terephthalic acid	Phenolic acids	Phenolic acids	0.06%	0.07%
Phthalic acid	Phenolic acids	Phenolic acids	0.01%	0.01%
Methyl gallate	Phenolic acids	Phenolic acids	0.95%	1.02%
Ethyl gallate	Phenolic acids	Phenolic acids	0.00%	0.00%
p-Coumaric acid	Phenolic acids	Phenolic acids	0.04%	0.04%
Neochlorogenic acid(5-*O*-caffeoylquinic acid)	Phenolic acids	Phenolic acids	0.09%	0.10%
Protocatechuic aldehyde	Phenolic acids	Phenolic acids	0.01%	0.01%
4-Methoxycinnamaldehyde	Phenolic acids	Phenolic acids	0.01%	0.01%
Oxalic acid	Phenolic acids	Phenolic acids	0.03%	0.01%
Protocatechuic acid-4-glucoside	Phenolic acids	Phenolic acids	0.02%	0.21%
Isochlorogenic acid A	Phenolic acids	Phenolic acids	0.11%	0.08%
Isochlorogenic acid C	Phenolic acids	Phenolic acids	0.13%	0.11%
1-*O*-[(E)-*p*-Cumaroyl]-β-D-glucopyranose	Phenolic acids	Phenolic acids	0.05%	0.55%
3-*O*-(E)-*p*-Coumaroyl quinic acid	Phenolic acids	Phenolic acids	0.04%	0.03%
3-Galloylshikimic acid	Phenolic acids	Phenolic acids	0.01%	0.01%
1-*O*-Galloyl-β-D-glucose	Phenolic acids	Phenolic acids	0.08%	0.12%
Galloyl methyl gallate	Phenolic acids	Phenolic acids	0.12%	0.14%
1,6-Bis-*O*-galloyl-β-D-glucose	Phenolic acids	Phenolic acids	0.30%	0.33%
Methyl 5-galloyl gallate	Phenolic acids	Phenolic acids	0.00%	0.00%
Esculetin	Phenolic acids	Phenolic acids	0.02%	0.00%
Gentisic acid	Phenolic acids	Phenolic acids	1.23%	0.87%
Hexadecanoic acid	Phenolic acids	Phenolic acids	2.80%	2.62%
Hexahydroxydiphenoyl galloylglucose	Phenolic acids	Phenolic acids	0.38%	0.37%
Glucogallin	Phenolic acids	Phenolic acids	0.31%	0.37%
Hexahydroxydiphenoylglucose	Phenolic acids	Phenolic acids	0.21%	0.17%
Digalloylglucose	Phenolic acids	Phenolic acids	0.55%	0.48%
Trihydroxycinnamoylquinic acid	Phenolic acids	Phenolic acids	0.01%	0.01%
Rosmarinyl glucoside	Phenolic acids	Phenolic acids	0.00%	0.00%
Oleoside 11-methyl ester	Phenolic acids	Phenolic acids	0.00%	0.00%
Trans-3-*O*-*p*-coumaric quinic acid	Phenolic acids	Phenolic acids	0.43%	0.40%
Chlorogenic acid	Phenolic acids	Phenolic acids	0.62%	0.59%
Phthalic anhydride	Phenolic acids	Phenolic acids	0.04%	0.05%
Procyanidin B1	Tannins	Proanthocyanidins	0.02%	0.05%
Theaflavin	Tannins	Proanthocyanidins	0.28%	0.07%
Theaflavin-3-gallate	Tannins	Proanthocyanidins	0.20%	0.03%
Theaflavin-3′-gallate	Tannins	Proanthocyanidins	0.18%	0.03%
Theaflavin 3,3′-digallate	Tannins	Proanthocyanidins	0.28%	0.05%
Procyanidin B2	Tannins	Proanthocyanidins	0.08%	0.10%
Procyanidin C1	Tannins	Proanthocyanidins	0.02%	0.02%
Procyanidin B4	Tannins	Proanthocyanidins	0.03%	0.06%
Procyanidin B3	Tannins	Proanthocyanidins	0.37%	0.45%
Procyanidin C2	Tannins	Proanthocyanidins	0.04%	0.05%
Ellagic acid	Tannins	Tannin	0.01%	0.01%
Cinnamtannin B2	Tannins	Tannin	0.00%	0.01%
Vomifoliol	Terpenoids	Monoterpenoids	0.00%	0.00%
Ribitol	Others	SA 7	0.01%	0.01%
D-Sorbitol	Others	SA	0.01%	0.01%
D-(+)-Trehalose anhydrous	Others	SA	0.06%	0.07%
D-Xylonic acid	Others	SA	0.24%	0.25%
D-Arabitol	Others	SA	0.01%	0.01%
L-Arabitol	Others	SA	0.01%	0.01%
Galactinol	Others	SA	0.48%	0.50%
Glucose-1-phosphate	Others	SA	0.09%	0.09%
Mannitol	Others	SA	0.00%	0.00%
Melibiose	Others	SA	0.07%	0.08%
Panose	Others	SA	0.00%	0.00%
D-Pinitol	Others	SA	0.01%	0.01%
Trehalose 6-phosphate	Others	SA	0.00%	0.00%
N-Acetyl-D-galactosamine	Others	SA	0.07%	0.06%
D-Glucose	Others	SA	0.20%	0.10%
D-Glucurono-6,3-lactone	Others	SA	0.00%	0.00%
Isomaltulose	Others	SA	0.15%	0.18%
Turanose	Others	SA	0.00%	0.00%
Glucarate O-phosphoric acid	Others	SA	0.07%	0.07%
D-(+)-Melezitose	Others	SA	0.00%	0.00%
Xylitol	Others	SA	0.01%	0.01%
Inositol	Others	SA	0.03%	0.03%
D-(+)-Sucrose	Others	SA	0.28%	0.29%
Gluconic acid	Others	SA	0.05%	0.05%
Pantothenol	Others	SA	0.00%	0.00%
DL-Arabinose	Others	SA	0.02%	0.02%
Dulcitol	Others	SA	0.01%	0.01%
D-Glucoronic acid	Others	SA	0.02%	0.01%
Resveratrol	Others	Stilbene	0.01%	0.00%
Nicotinamide	Others	Vitamin	1.79%	1.86%
Riboflavin	Others	Vitamin	0.06%	0.04%
Pyridoxal 5′-phosphate	Others	Vitamin	0.00%	0.00%
D-Pantothenic acid	Others	Vitamin	0.31%	0.36%
Nicotinic acid	Others	Vitamin	0.01%	0.01%
Pyridoxine	Others	Vitamin	0.12%	0.06%
Biotin	Others	Vitamin	0.00%	0.00%
4-Pyridoxic acid	Others	Vitamin	0.01%	0.01%
Maltol	Others	Others	0.01%	0.01%

^1^ AAD: amino acids and derivatives, ^2^ LC: lignans and coumarins, ^3^ LPC: lysophosphatidylcholine, ^4^ LPE: lysophosphatidylethanolamine, ^5^ PC: phosphatidyl cholines, ^6^ ND: nucleotides and derivatives, ^7^ SA: saccharides and alcohols.

## Data Availability

The experimental data provided in this work are available in articles and Appendix A.

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
