# Peer review of "An In Vitro Catalysis of Tea Polyphenols by Polyphenol Oxidase"

_molecules, 2023, doi:10.3390/molecules28041722_

Round 1
Reviewer 1 Report
1. In abstract, the authors do not make clear what these numbers mean, such as flavonoids (125), phenolic acids (67).
2. All abbreviations must be provided full names at the first appearance, such as EGGC, EGC and so on.
3. In Fig,1, what's the mean of superscript (a, b, c, d)
4. In Fig. 2, there is no peak at the position of Rt and Qc. In contrast, there is one small peak between GCG and ECG, but the authors did not name it.
Author Response
Dear Reviewer:
Thank you for your comments concerning our manuscript entitled “An In Vitro Catalysis of Tea Polyphenols by Polyphenol Oxidase”. Those comments are all valuable and very helpful for revising and improving our paper, as well as the important guiding significance to our researches. We have studied comments carefully and have made correction which we hope meet with approval. And we checked the English writing carefully, and it was improved by editors from International Science Editing (http://www.internationalscienceediting.com), as well as. We hope this improvement can be accepted. The main corrections in the paper and the responds to your comments are as flowing:
- Comment: In abstract, the authors do not make clear what these numbers mean, such as flavonoids (125), phenolic acids (67).
Response: This suggestion is very important, we appreciate this suggestion and it has been revised to “flavonoids (125 metabolites), phenolic acids (67 metabolites), lipids (55 metabolites)”.
- Comment:All abbreviations must be provided full names at the first appearance, such as EGGC, EGC and so on.
Response: We agree with this suggestion and it has been revised.
- Comment: In Fig,1, what's the mean of superscript (a, b, c, d).
Response: We thank this question, and different lowercase superscripts within a row indicated significantly different among comparisons (P < 0.05), please see Figure 1 in revision.
- Comment: In Fig. 2, there is no peak at the position of Rt and Qc. In contrast, there is one small peak between GCG and ECG, but the authors did not name it.
Response: This suggestion is very important, we appreciate this suggestion. It has been checked and revised, and the nameless peak is not tea polyphenols.
Once again, thank you very much for your comments and suggestions.
Yours sincerely,
Kunyi Liu
Reviewer 2 Report
Review Report
Manuscript ID: molecules-2135802
Title: An In Vitro Catalysis of Tea Polyphenols by Polyphenol Oxidase
This manuscript deals with an in vitro catalysis of tea polyphenols from sun-dried green tea which was previously grinding to a fine powder by liquid nitrogen. The polyphenol oxidase catalytic ability of tea polyphenols was analyzed and optimized. Authors have identified 441 metabolites in the enzymatic tea powder which were classified into 11 classes.
Reviewer’s observations and suggestions
An abstract is correctly written.
An introduction is very informative regarding the aims, with up-to-date references.
Results and discussion.
The authors can give a little bit deeper explanation of the clustering presented in a Fig. 3A.
Materials and Methods
Lines 188-189 Give some more details about powdering with liquid N2 "Sun-dried green tea leaves were ground to a fine powder with liquid nitrogen to obtain tea leaves with broken cell walls."
This study has a high potential to be cited in large.
I highly recommend that the Editorial Office consider this manuscript for publication after minor revision.
Author Response
Dear Reviewer:
Thank you for your comments concerning our manuscript entitled “An In Vitro Catalysis of Tea Polyphenols by Polyphenol Oxidase”. Those comments are all valuable and very helpful for revising and improving our paper, as well as the important guiding significance to our researches. We have studied comments carefully and have made correction which we hope meet with approval. The main corrections in the paper and the responds to your comments are as flowing:
- Comment: The authors can give a little bit deeper explanation of the clustering presented in a Fig. 3A.
Response: This suggestion is very important, we appreciate this suggestion and it has been revised.
- Comment:Lines 188-189 Give some more details about powdering with liquid N2 "Sun-dried green tea leaves were ground to a fine powder with liquid nitrogen to obtain tea leaves with broken cell walls."
Response: We agree with this suggestion and it has been revised.
Once again, thank you very much for your comments and suggestions.
Yours sincerely,
Kunyi Liu
Reviewer 3 Report
This study used the commercial polyphenol oxidase to catalysis the tea powder, and analyzed the metabolites, then compared their variations. Although this manuscript gave some data about these items, but it seemed lack of highlights, due to present many literature on studies of tea polyphenols oxidation by polyphenol oxidase.
Even this manuscript found 441 metabolites, only a few of them was meaningful, and it was still very difficult to get useful information directly from those words in the text.
Fig.5 might have a confused composition? The formation mechanism for theaflavins here was no meaningful, and it was very common knowledge for researchers in tea science area.
Author Response
Dear Reviewer:
Thank you for your comments concerning our manuscript entitled “An In Vitro Catalysis of Tea Polyphenols by Polyphenol Oxidase”. Those comments are all valuable and very helpful for revising and improving our paper, as well as the important guiding significance to our researches. We have studied comments carefully and have made correction which we hope meet with approval. And we checked the English writing carefully, and it was improved by editors from International Science Editing (http://www.internationalscienceediting.com), as well as. We hope this improvement can be accepted. The main corrections in the paper and the responds to your comments are as flowing:
- Comment: this manuscript found 441 metabolites, only a few of them was meaningful, and it was still very difficult to get useful information directly from those words inthe text.
Response: We thank this question. All the differentially changed tea polyphenols (VIP > 1.0, P < 0.05 and FC > 2 or < 0.5) have been listed in the manuscript, while it is proved that 29 tea polyphenols (e.g., theaflavin, theaflavin-3'-gallate, theaflavin-3-gallate, theaflavin 3,3'-digallate, etc.) can be produced by polyphenol oxidase (PPO).
- Comment: Fig.5 might have a confused composition? The formation mechanism for theaflavins here was no meaningful, and it was very common knowledge for researchers in tea science area.
Response: This suggestion is very important and it has been revised. As showed in Fig. 5, comparison with CK, the levels of EC, EGC, ECG, EGCG in CTP decreased 76.87%, 78.16%, 65.63% and 71.98%, respectively, whereas the levels of TF1, TF2A, TF2B, TF3 increased 3.82-, 5.11-, 5.92- and 6.01-fold, respectively. We suggested that TF1 was synthesized through the polymerization of EC and EGC under the catalysis of PPO, PPO could catalyze the polymerization of EC and EGCG to form TF2A, ECG and EGC could be polymerized to form TF2B under the catalysis of PPO, and TF3 was synthesized through the polymerization of ECG and EGCG under the catalysis of PPO. Therefore, Fig. 5 showed that the changes in the relative levels of the substrate and the product under the catalysis of polyphenol oxidase, and that TF1, TF2A, TF2B, TF3 can be produced by enzymatic oxidation only in the presence of dihydroxy-B-cycloflavanol (e.g., EC and ECG) and trihydroxy-B-cycloflavanol (e.g., EGC and EGCG) through the structural formula and the change of the levels of metabolites.
Once again, thank you very much for your comments and suggestions.
Yours sincerely,
Kunyi Liu
Round 2
Reviewer 3 Report
The manuscript has been revised and interpretated carefully.
I think it might be accepted after next type-writing check.
Author Response
Dear Reviewer:
Thank you for your comments concerning our manuscript entitled “An In Vitro Catalysis of Tea Polyphenols by Polyphenol Oxidase”. We have studied comments carefully and have made correction which we hope meet with approval. And we checked the English writing carefully, and it was improved by editors from International Science Editing (http://www.internationalscienceediting.com), as well as.